# A Valence-Bond-Based Multiconfigurational Density Functional Theory: The λ-DFVB Method Revisited

**DOI:** 10.3390/molecules26030521

**Published:** 2021-01-20

**Authors:** Peikun Zheng, Chenru Ji, Fuming Ying, Peifeng Su, Wei Wu

**Affiliations:** Fujian Provincial Key Laboratory of Theoretical and Computational Chemistry, The State Key Laboratory of Physical Chemistry of Solid Surfaces, College of Chemistry and Chemical Engineering, Xiamen University, Xiamen 361005, China; pkzheng@stu.xmu.edu.cn (P.Z.); jichenru@stu.xmu.edu.cn (C.J.); fmying@xmu.edu.cn (F.Y.)

**Keywords:** valence bond theory, density functional theory, electron correlation, multireference

## Abstract

A recently developed valence-bond-based multireference density functional theory, named λ-DFVB, is revisited in this paper. λ-DFVB remedies the double-counting error of electron correlation by decomposing the electron–electron interactions into the wave function term and density functional term with a variable parameter λ. The λ value is defined as a function of the free valence index in our previous scheme, denoted as λ-DFVB(K) in this paper. Here we revisit the λ-DFVB method and present a new scheme based on natural orbital occupation numbers (NOONs) for parameter λ, named λ-DFVB(IS), to simplify the process of λ-DFVB calculation. In λ-DFVB(IS), the parameter λ is defined as a function of NOONs, which are straightforwardly determined from the many-electron wave function of the molecule. Furthermore, λ-DFVB(IS) does not involve further self-consistent field calculation after performing the valence bond self-consistent field (VBSCF) calculation, and thus, the computational effort in λ-DFVB(IS) is approximately the same as the VBSCF method, greatly reduced from λ-DFVB(K). The performance of λ-DFVB(IS) was investigated on a broader range of molecular properties, including equilibrium bond lengths and dissociation energies, atomization energies, atomic excitation energies, and chemical reaction barriers. The computational results show that λ-DFVB(IS) is more robust without losing accuracy and comparable in accuracy to high-level multireference wave function methods, such as CASPT2.

## 1. Introduction

The electronic structure calculations of strongly correlated systems, which cannot be well described by a single configuration space function, are still challenging in the methodology development of electronic structure theory. Typical strongly correlated problems occur in the excited states; transition states; open-shell molecules, especially the radical species and systems that contain transition-metal atoms; and the bond dissociation process, etc. Obtaining a proper description of such systems usually requires a method capable of describing both static and dynamic electron correlations.

The complete active space self-consistent field (CASSCF) [1] is a widely used quantum chemistry method able to capture static correlation. In valence bond (VB) theory, the valence bond self-consistent field (VBSCF) [2,3], which is a multiconfigurational self-consistent field (MCSCF) analog with atomic orbitals (AOs), covers the static correlation by expressing the many-electron wave function of the molecule as a linear combination of VB structures. To cover dynamic correlations within the CASSCF scheme, post-CASSCF methods are required, such as the complete active space second-order perturbation theory (CASPT2) [4] and multireference configuration interaction (MRCI) [5]. Their analogs are present in VB theory, valence bond configuration interaction (VBCI) [6,7], and valence bond perturbation theory (VBPT2) [8,9], respectively. However, the computational costs of both methods are still expensive, compared to their MO analogs. 

Kohn−Sham density functional theory (KS-DFT) [10,11] has played a very important role in electronic structure calculation in chemistry, biochemistry, and solid state due to its affordable computational cost with satisfactory accuracy. Due to the fact that the exact formula of the exchange–correlation functional is still unknown, KS-DFT currently suffers from a difficulty in tackling molecular systems with a strong multireference character, which cannot be described well by using a single Slater determinant. One of the straightforward schemes is to combine CASSCF with DFT, namely multireference density functional theory (MRDFT), where CASSCF takes care of the static correlation, while DFT describes the dynamic correlation. Various MRDFT schemes have been proposed to properly describe strongly correlated systems [12,13,14,15,16,17,18,19,20,21,22,23,24,25,26,27,28,29]. One of the problems that should be solved in MRDFT is the double-counting error (DCE), which is due to the fact that static and dynamic correlations cannot be separated strictly. By using on-top pair density, various DCE-free MRDFT methods have been proposed over the last two decades [25,30,31,32,33,34]. Among them, one of the representative MRDFT methods is multiconfiguration pair-density functional theory (MC-PDFT), which avoids the DCE by calculating the kinetic and Coulomb contributions of the wave function part, and the rest of the part is obtained from a pair-density functional [25].

In a similar fashion, the strategy of hybrid VB theory with DFT has been implemented, such as VBDFT(s) [35,36,37,38], VB-DFT [39], and DFVB [40,41,42]. Recently, a newly developed MRDFT scheme based on the valence bond wave function, namely the λ-density functional valence bond (λ-DFVB) method [42], was presented. This method is inspired by the MC1H approximation proposed by Sharkas et al. [22]. The MC1H approach combines the MCSCF with DFT by linear decomposition of the electron−electron interaction with a single parameter λ, which is set to a fixed value of 0.25. Different from Sharkas’s scheme, the λ value in λ-DFVB is variable and ranges from 0 to 1, depending on the multireference character of the studied molecule. The stronger the multireference character, the larger value taken in the calculation. To describe the extent of the multireference character, the molecular free valence index *K* is used in λ-DFVB, and the functional form of λ is taken tentatively as λ = *K*^1/4^. The molecular free valence index *K* is defined by the total atomic free valences and the bond orders of the molecule, expressed in terms of the overlap and spin density matrices [42].

Estimates of the multireference character have been explored by various approaches. Among them, indices based on the natural orbital occupation numbers (NOONs) are widely used, which are given by diagonalizing the density matrix of the molecule. Various schemes have been proposed, such as von Neumann entropy *S* [43], correlation entropy *S*_2_ [44], the *M* diagnostic [45], and the *I_ND_* diagnostic [46]. The purpose of this paper is to revisit the λ-DFVB method by employing the NOONs to determine the value of parameter λ to simplify the procedure of λ-DFVB calculation. Different from the molecular free valence index, which is defined by the total density matrix and spin density matrix as well as orbital overlaps, NOONs are straightforwardly obtained from the density matrix of the studied molecule. Moreover, the concept of bond order used in λ-DFVB(K) is somehow ambiguous if one performs λ-DFVB calculation for atomic systems.

In this paper, we revisit the λ-DFVB method and present a new scheme based on NOONs for parameter λ, named λ-DFVB(IS), instead of the molecular free valence index *K* used in the previous λ-DFVB scheme, which is denoted as λ-DFVB(K). Furthermore, to improve the practicality of the λ-DFVB method, some technical considerations are investigated in this paper to further simplify the process of λ-DFVB calculation.

## 2. Methodology

In VB theory, a many-electron wave function is expanded as a linear combination of Heitler−London−Slater−Pauling (HLSP) functions [47,48],
(1)$$\Psi =\sum_{K}C_{K}\Phi_{K}$$
where Φ*_K_* is a specific VB structure, and *C_K_* is the corresponding coefficient. The HLSP function Φ*_K_* can be expressed as follows
(2)$$\Phi_{K}=\hat{A}\Omega_{0}\Theta_{K}$$
where $\hat{A}$ is the antisymmetrizing operator, and Ω_0_ is a direct product of orbitals {*ϕ*_i_} as
(3)$$\Omega_{0}=\phi_{1}\left(1\right)\phi_{2}\left(2\right)\cdots \phi_{N}\left(N\right)$$
and Θ*_K_* is a spin-paired spin eigenfunction [49], defined as
(4)$$\Theta_{K}=\prod_{\left(ij\right)}2^{-1/2}\left[\alpha \left(i\right)\beta \left(i\right)-\beta \left(i\right)\alpha \left(j\right)\right]\prod_{k}\alpha \left(k\right)$$
where (*ij*) runs over all bonds and *k* over all unpaired electrons.

The total energy and structure coefficients can be obtained by solving the standard secular equation
(5)HC=EMC
where **H**, **M**, and **C** are the VB Hamiltonian, overlap, and coefficient matrices, respectively. The weight of a given VB structure, *W**_K_*, can be evaluated by the Coulson–Chirgwin formula [50],
(6)$$W_{K}=\sum_{L}C_{K}C_{L}M_{KL}$$

As the wave function theory (WFT) of electronic structure, there are various levels in ab initio classical VB methods, such as the valence bond self-consistent field (VBSCF) [2,3], breathing orbital valence bond (BOVB) [51,52,53], valence bond configuration interaction (VBCI) [6,7], and valence bond second-order perturbation theory (VBPT2) [8,9]. The VBSCF method is the elementary method of ab initio classical VB theory, where both VB structure coefficients {*C_K_*} and VB orbitals {*ϕ*_i_} are optimized simultaneously to minimize the total energy.

The λ-DFVB method incorporates the dynamic energy into VB theory using KS-DFT by expressing the total electronic energy of a molecule as
(7)$$E^{\lambda -DFVB}=\underset{\Psi }{min}\left\{\langle \Psi |T+V_{ext}+\lambda W_{ee}\left|\Psi \right.\rangle +E_{HXC}^{\lambda -DFVB}[\rho ]\right\}$$
where *T*, *V*_ext_, and *W*_ee_ are the kinetic energy, external potential, and electron–electron interaction operators, respectively; $E_{HXC}^{\lambda -DFVB}[\rho ]$ is the complement λ-dependent Hartree–exchange–correlation density functional for electronic density *ρ*; and λ is a coupling parameter. The first term in Equation (7) is computed by a normal VB route, while the second term, $E_{HXC}^{\lambda -DFVB}[\rho ]$, is calculated by the KS-DFT method and includes four components
(8)$$E_{HXC}^{\lambda -DFVB}[\rho ]=(1-\lambda )\left(E_{H}[\rho ]+E_{X}[\rho ]\right)+\left(1-\lambda^{2}\right)E_{C}[\rho ]+\lambda^{2}E_{C}[\rho^{LD}]$$
where $E_{H}[\rho ]$ is the Hartree energy, $E_{X}[\rho ]$ is the exchange functional, and $E_{C}[\rho ]$ is the correlation functional. Different from the multiconfigurational one-parameter hybrid (MC1H) approach [22], an extra term of the correlation energy, $E_{C}[\rho^{LD}]$, is included, which is the correlation functional for the determinant that shares the largest coefficient in the VBSCF wave function.

As discussed in the previous paper, the parameter λ in Equation (7) scales the hybrid extent of the WFT and the KS-DFT method. Based on the fact that the VBSCF method covers the static correlation, it is thus suitable for molecules with a multireference character, while KS-DFT is a good tool for capturing the dynamic correlation. As such, it is more reasonable to allow the value of λ to be different for molecules with different extents of static and dynamic correlations.

In λ-DFVB(K), parameter λ is defined as a function of the molecular free valence index *K*,
(9)$$\lambda =K^{1/4}$$

The molecular free valence index *K* is defined as
(10)$$K=\frac{\sum_{A}F_{A}}{\sum_{A}V_{A}}$$
(11)$$F_{A}=V_{A}-\sum_{B,B\neq A}O_{AB}$$
(12)$$V_{A}=\sum_{\mu \in A}2{\left(DS\right)}_{\mu \mu }-\sum_{\mu ,\nu \in A}{\left(DS\right)}_{\mu \nu }{\left(DS\right)}_{\nu \mu }$$
(13)$$O_{AB}=\sum_{\mu \in A}\sum_{\nu \in B}\left[{\left(DS\right)}_{\mu \nu }{\left(DS\right)}_{\nu \mu }+{\left(P^{s}S\right)}_{\mu \nu }{\left(P^{s}S\right)}_{\nu \mu }\right]$$
where *F*_A_ and *V*_A_ are the free valence and total valence of atom A [54], *O*_AB_ is the bond order between atoms A and B, **S** is the overlap matrix between the basis functions, and **D** and **P**^s^ are the total and spin polarization density matrices, respectively. The value of *K* ranges from 0 to 1, as *F*_A_ is identical to *V*_A_ in the dissociation limit.

Equation (7) can be implemented by a modified VBSCF route [42]. To this end, the structure coefficients {*C_K_*} and orbitals {*ϕ_i_*} are optimized by minimizing
(14)$$\varepsilon_{}^{\lambda -DFVB}=\langle \Psi |T+V_{ext}+\lambda W_{ee}+\int drv_{HXC}^{\lambda -DFVB}\left[\rho \right]n\left(r\right)|\Psi \rangle$$
where $n\left(r\right)$ is the density operator, and potential $v_{HXC}^{\lambda -DFVB}\left[\rho \right]$ is defined as
(15)$$v_{HXC}^{\lambda -DFVB}\left[\rho \right]=\frac{\delta E_{HXC}^{\lambda -DFVB}\left[\rho \right]}{\delta \rho }$$

In implementation, the potential $v_{HXC}^{\lambda -DFVB}\left[\rho \right]$ is expressed in terms of one-electron integrals and λ-scaled two-electron integrals of the basis functions. The total electronic energy of the molecule is computed with the optimized {*C_K_*} and {*ϕ_i_*}, which is expressed as
(16)$$E_{}^{\lambda -DFVB}=\varepsilon_{}^{\lambda -DFVB}+E_{HXC}^{\lambda -DFVB}\left[\rho \right]-\langle \Psi |\int drv_{HXC}^{\lambda -DFVB}\left[\rho \right]n\left(r\right)|\Psi \rangle$$

Before performing a modified VBSCF calculation, a normal VBSCF is done for determining the value of parameter λ. Thus, the λ-DFVB(K) calculation is composed of the following steps:
(1)Perform a normal VBSCF calculation to obtain the VBSCF density matrix;(2)Determine the λ value with Equations (9)–(13);(3)Compute the complement Hartree–exchange–correlation functional, $E_{HXC}^{\lambda -DFVB}[\rho ]$ with Equation (8);(4)Build potential $v_{HXC}^{\lambda -DFVB}\left[\rho \right]$ in terms of one-electron integrals with Equation (15);(5)Optimize {*C_K_*} and {*ϕ_i_*} by a modified VBSCF route, Equation (14);(6)Compute the λ-DFVB energy by Equation (16) with the optimized {*C_K_*} and {*ϕ_i_*}.

As can be seen above, the most time consuming occurs in Steps (1) and (5), which involve SCF iterations. Thus, the computational effort for λ-DFVB(K) is approximately two times that of VBSCF.

To determine the value of parameter λ in λ-DFVB(K), one should compute the free valence *F*_A_ and total valence *V*_A_, as well as the bond order *O*_AB_, for all atoms, which are defined with the density and overlap matrices of the basis functions. It can be shown that for closed-shell molecules, the summation of free valence *F*_A_ over all atoms actually equals the number of effectively unpaired electrons (EUEs), *N*_D_,
(17)$$\sum_{A}F_{A}=N_{D}=\sum_{i}n_{i}\left(2-n_{i}\right)$$
where *n_i_* is the natural orbital occupation numbers. It is worth noting that the number of EUEs and their variants have been widely used for estimating the static correlation character of a molecule [46,55,56,57,58,59]. To simplify the definition of parameter λ, the number of EUEs is used in this paper, instead of *K*. To ensure that the value of the index lies between 0 and 1, a normalized index based on EUEs is defined as
(18)$$I_{s}=\frac{N_{D}}{2n-n^{2}/m}$$
where *n* and *m* are the numbers of active electrons and active orbitals, respectively. The details of deduction for Equation (17) and the normalization factor in Equation (18) are presented in Appendix B.

In a similar fashion to λ-DFVB(K), parameter λ is defined by a function of *I_s_*, as
(19)$$\lambda =I_{s}{}^{1/4}$$

Obviously, the range of the λ value is from 0 to 1, the same as *I_s_*. Thus, the new scheme of λ-DFVB is denoted as λ-DFVB(IS). Now index *I_s_* depends only on the NOONs, which are given by diagonalizing the density matrix of the molecule. Clearly, λ-DFVB(IS) gets rid of a set of intermediate quantities, such as free valence, total valence, and bond order, making the scheme more concise.

Furthermore, the numerical investigation shows that the optimized orbitals given from Step (1) in λ-DFVB(K) are virtually the same as the orbitals optimized in Step (5). As such, in λ-DFVB(IS), Step (5) is skipped, and only one SCF iteration process, Step (1), is required. That is to say, λ-DFVB(IS) is a post-VBSCF method, where VB orbitals and the density matrix are given at the VBSCF level, and the Hartree–exchange–correlation density functional is computed with the VBSCF density. Thus, the λ-DFVB(IS) energy is expressed as
(20)$$E_{}^{\lambda -DFVB}=E^{VBSCF}\left(\lambda \right)+E_{HXC}^{\lambda -DFVB}\left[\rho \right]$$
where *E*^VBSCF^(λ) is VBSCF energy with the λ-dependent scaled two-electron integrals.

The implementation of λ-DFVB(IS) includes the following four steps:
(1)Perform a normal VBSCF calculation to obtain the VBSCF density matrix;(2)Compute NOONs {*n_i_*} from the VBSCF wave function, and get the λ value using Equation (19);(3)Compute the complement Hartree–exchange–correlation functional, $E_{HXC}^{\lambda -DFVB}[\rho ]$ with Equation (8);(4)Compute the λ-DFVB(IS) energy by Equation (20).

It is obvious that the current scheme, λ-DFVB (IS), is much simpler than the old scheme, λ-DFVB(K).

## 3. Computational Details

All the VB calculations were implemented in the Xiamen Valence Bond (XMVB) package [60,61] using full VB structures and overlap-enhanced orbitals (OEOs), while the KS-DFT calculation were performed by the Gaussian 16 program [62]. The BLYP [63,64] functional was used for all λ-DFVB calculations. For comparison, CASPT2 calculations were also performed using OpenMolcas [65], with a standard imaginary shift of 0.2 Hartrees and the default IPEA shift of 0.25 Hartrees.

Test calculations can be classified into four sets: potential energy curves, atomization energies, excitation energies, and reaction barrier heights. For the potential energy curves, some diatomic molecules were investigated, including H_2_, N_2_, C_2_, F_2_, and HF. To assess the performance of atomization energies, the AE6 dataset [66] was used, which consists of SiH_4_, S_2_, SiO, C_3_H_4_, C_2_H_2_O_2_, and C_4_H_8_. The excitation energies of several main group atoms, Be, C, N, N^+^, O, and O^+^, were also investigated. Finally, a series of reaction barriers in the DBH24 dataset [67] were studied, which contains 12 reactions with both forward and reverse reaction barrier heights.

The basis set used for the potential energy curves, atomization energies, and excitation energies was the cc-pVTZ basis set [68], while for the reaction barriers, the maug-cc-pVTZ basis set [69] was used. The geometries for the molecules in the AE6 and DBH24 datasets were taken from the Minnesota Database 2019 [70].

## 4. Results and Discussion

### 4.1. The Validity of the N_D_ Index

In order to validate the use of *N*_D_ in measuring the static correlation, Figure 1 shows the plots of the correlation entropy $S_{2}=-\sum_{i}\left(n_{i}/2\right)ln\left(n_{i}/2\right)$, which is widely used for diagnosing the extent of the multireference character versus the free valence index *K* (a) and the number of effectively unpaired electrons, the *N*_D_ index, (b) for the transition states in the DBH24 datasets computed by VBSCF. As can be seen from Figure 1, index *N*_D_ shares a stronger correlation with *S*_2_ than index *K*. The value of *R*^2^ for *S*_2_ vs. *N*_D_ is 0.9399, and it is 0.3233 for *S*_2_ vs. *K*. Figure 1 shows that it may be more reliable to use index *N*_D_ for the λ-DFVB method, instead of index *K*.

### 4.2. Dissociation of Diatomic Molecules

The results of the equilibrium bond lengths and bond dissociation energies calculated by various methods are presented in Table 1 and Table 2, respectively. For comparison, the deviation values from the reference values are listed. The mean unsigned error (MUE) is also listed in the bottom row.

It can be seen in Table 1 that among all the computational methods, CASPT2 performs the best, followed by B3LYP. The equilibrium bond lengths predicted by the two schemes of the λ-DFVB method show a good agreement, with a MUE value of 0.012 Å for λ-DFVB(K) and 0.011 Å for λ-DFVB(IS), both of which improved from the value of 0.017 Å of VBSCF. Both λ-DFVB(K) and λ-DFVB(IS) predicted shorter equilibrium bond lengths for all molecules, compared to VBSCF. This makes sense as VBSCF usually provides a slightly longer bond length due to the lack of a dynamic correlation. For all methods, the largest deviation comes from the F_2_ molecule. For example, the deviation of *R*_e_ predicted by VBSCF is ca 0.055 Å, whereas the corresponding deviations are −0.040 Å for λ-DFVB(K) and −0.035 Å for λ-DFVB(IS).

It can be seen from Table 2 that as expected, the largest MUE for bond dissociation energy (BDE), 18.6 kcal/mol, comes from VBSCF. The MUE of λ-DFVB(K), 4.5 kcal/mol, is much improved from VBSCF, while the MUE of 1.9 kcal/mol of λ-DFVB(IS) is even better, which is the best over all the methods. For molecules with a single bond, H_2_, F_2_, and HF, the deviations of *D*_e_ of the two λ-DFVB methods are smaller than 2 kcal/mol. For molecules with multiple bonds, N_2_ and C_2_, the deviations are a little larger for λ-DFVB(K), −3.1 kcal/mol for N_2_ and −15.9 kcal/mol for C_2_. For λ-DFVB(IS), the BDE value of N_2_ is somewhat lower than λ-DFVB(K), while the best MUE of C_2_ comes from λ-DFVB(IS). From the potential energy curves, shown in the Appendix A, we can find that the two schemes both give correct dissociation behavior, while the curves of λ-DFVB(IS) locate more closely to the high-level CASPT2 than λ-DFVB(K).

### 4.3. Atomization Energies of Six Molecules

Table 3 collects the atomization energies of six molecules in the AE6 dataset, which are divided by the number of bonds, and the MUE values are calculated using the converted values, similar to [74]. As expected, VBSCF has the largest MUE value of 14.9 kcal/mol due to the lack of dynamic correlation energy. The MUE errors of the two schemes of λ-DFVB are 3.1 kcal/mol and 3.7 kcal/mol, respectively, in good agreement with that of CASPT2 (MUE = 3.2 kcal/mol), compared to the MUE value of MC-PDFT, about 2.3 kcal/mol [74]. KS-DFT provides better MUE values, 1.2 kcal/mol and 2.0 kcal/mol, for BLYP and B3LYP, respectively. The largest deviation comes from the *S*_2_ molecule, 10.3 kcal/mol and 9.3 kcal/mol, respectively, for λ-DFVB(K) and λ-DFVB(IS), and CASPT2 also has an error of about 5 kcal/mol. This may be due to its triplet character, which requires a balanced treatment of static and dynamic correlation. This can also be confirmed by the results shown in Appendix A. All the indexes and λ share the largest values for this molecule.

### 4.4. Atomic Excitation Energies

The excitation energies between different spins of several atoms in the second row are collected in Table 4. It can be found that CASPT2 predicts the excitation energies fairly well with respect to the reference values, with a MUE of only 0.05 eV, followed by λ-DFVB(IS), with a MUE of 0.2 eV. As expected, the excitation energies calculated by KS-DFT share larger MUE values, 1.2 eV for BLYP and 1.14 eV for B3LYP. On the contrary, VBSCF tends to overestimate the excitation energies, with an average error of 0.27 eV, but much better than the KS-DFT results. It is interesting that for all atoms, λ-DFVB(IS) markedly improves the atomic excitation energy compared to the results of λ-DFVB(K), giving an improvement in the MUE of 0.12 eV.

As mentioned above, the concept of bond order is ambiguous when dealing with atomic systems. From Appendix A, we can find that all the values of index *K* are equal to one for all the atomic systems because the calculation of free valences involves bond orders, which has been shown in Equation (11). This may account for the slightly worse result obtained by λ-DFVB(K) compared to VBSCF. However, the new index *I_s_* based on NOONs overcomes this problem. It can distinguish the extent of the multireference character well for each atomic system and state, which leads to the consistent improvement over all atomic excitation energies. 

### 4.5. Chemical Reaction Barriers

Chemical reaction barriers are challenging for electronic structure methods. A balanced description of both static and dynamic correlations is required. Quantitatively predicting the reaction barrier height generally requires predicting not only the forward reaction barrier but also the reverse reaction barrier accurately. The proper description of transition states is very critical because there sometimes exist near-degeneracy effects in the transition states, and multireference methods are usually needed.

Table 5 gives the forward and reverse barriers computed by CASPT2, VBSCF, λ-DFVB(K), and λ-DFVB(IS), alongside the results of BLYP and B3LYP. As shown in Table 5, VBSCF fails to provide satisfactory performance in predicting the reaction barriers with a slightly larger MUE value of 10.6 kcal/mol, and for most reactions, VBSCF predicts much higher barrier heights compared to the reference values. The CASTP2 results are in good agreement with the reference data, with the lowest MUE of 1.4 kcal/mol, which indicates the importance of introducing dynamic correlation. It is encouraging to note that λ-DFVB(K) predicts the correct sign for all the forward or reverse barriers, with an accuracy very similar to CASPT2, about 2.6 kcal/mol, similar to the MC-PDFT result, 3.2 kcal/mol [74]. λ-DFVB(IS) also obtains the correct sign for most of the reactions, except for the forward barrier of Reaction 9, which is overestimated by 3.6 kcal/mol. As for BLYP and B3LYP, both are inclined to underestimate the reaction barriers. The average error of BLYP is the highest (7.7 kcal/mol) of all methods, and by using the hybrid functional B3LYP, the MUE error can be reduced to about 4.2 kcal/mol, showing that the inclusion of exact exchange energy in functional is helpful for describing chemical reaction barriers to some extent.

## 5. Conclusions

In this paper, we have revisited a valence-bond-based multiconfigurational density functional theory, called λ-DFVB, which applies a single variable, parameter λ, to the decomposition of the electron repulsion operator. Different from the previous work, λ-DFVB(K), where parameter λ was determined by the free valence of a molecule, in this paper, we present a simplified definition for parameter λ. The value of parameter λ is given by the natural orbital occupation numbers, which are straightforwardly computed by diagonalizing the density matrix, and the new scheme is called λ-DFVB(IS). Moreover, the new DFVB method simplifies the computing process by omitting the iterative VBSCF calculation with the exchange–correlation functional potential, which sometimes results in the convergence problem. Thus, the computational effort in λ-DFVB(IS) is approximately the same as the VBSCF method, greatly reduced from λ-DFVB(K).

λ-DFVB(IS) is validated with the various physical and chemical properties of molecules, potential energy curves, atomization energies, excitation energies, and reaction barrier heights. The test results show that the performance of λ-DFVB(IS) is similar to λ-DFVB(K), and both of them share approximately the same accuracy as CASPT2. λ-DFVB(IS) is improved to some extent by λ-DFVB(K) in the dissociation energies and excitation energies. Owing to the simplification of the computational process, the CPU time used in λ-DFVB(IS) is much reduced, less than half of λ-DFVB(K).

In conclusion, λ-DFVB(IS) provides a simpler and cheaper hybrid method of valence bond theory and density functional theory, compared to λ-DFVB(K). The tests performed in this paper validate that λ-DFVB can serve as an electronic structure tool for strongly correlated systems, where current KS-DFT functionals are not able to provide satisfactory performance.

## Figures and Tables

**Figure 1 molecules-26-00521-f001:**
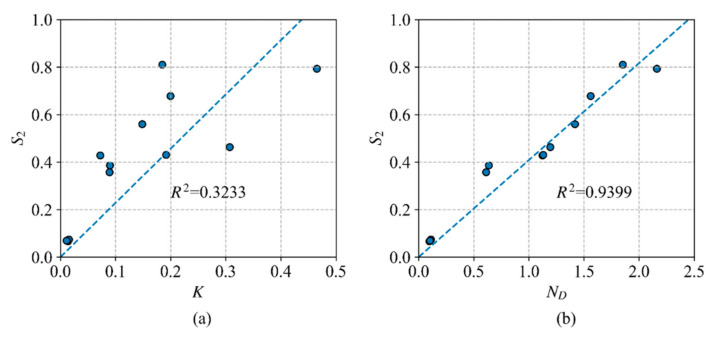
Plots of the correlation entropy *S*_2_ versus *K* (**a**) and *N*_D_ (**b**) for the transition states in the DBH24 datasets.

**Table 1 molecules-26-00521-t001:** Deviation values of the equilibrium bond lengths (*R*_e_, in Å) of diatomic molecules.

	Active Space	CASPT2	VBSCF	λ-DFVB		BLYP	B3LYP	Ref. ^1^
				K	IS			
H_2_	(2, 2)	0.004	0.014	0.003	0.003	0.005	0.002	0.741
F_2_	(2, 2)	0.007	0.055	−0.040	−0.035	0.020	−0.015	1.412
HF	(2, 2)	−0.001	−0.001	−0.009	−0.008	0.016	0.005	0.917
N_2_	(6, 6)	0.005	0.005	0.000	−0.003	0.005	−0.007	1.098
C_2_	(8, 8)	0.007	0.012	0.008	0.008	0.013	0.004	1.243
	MUE	0.005	0.017	0.012	0.011	0.012	0.007	

^1^ The reference values are taken from [71].

**Table 2 molecules-26-00521-t002:** Deviation values of the bond dissociation energies (*D*_e_, in kcal/mol) of diatomic molecules.

	Active Space	CASPT2	VBSCF	λ-DFVB		BLYP	B3LYP	Ref. ^1^
				K	IS			
H_2_	(2, 2)	−3.6	−14.2	−1.0	−1.1	0.0	0.7	109.5
F_2_	(2, 2)	−3.2	−21.4	1.0	0.2	12.4	−0.1	38.2
HF	(2, 2)	−4.1	−27.7	1.6	−0.1	−4.0	−4.2	141.3
N_2_	(6, 6)	−11.7	−24.4	−3.1	−6.5	11.3	0.5	228.5
C_2_	(8, 8)	−1.9	−5.4	−15.9	−1.6	−12.7	−28.5	148.0
	MUE	4.9	18.6	4.5	1.9	8.1	6.8	

^1^ The reference values of H_2_, F_2_, HF, and N_2_ are taken from [72], and that of C_2_ is from [73].

**Table 3 molecules-26-00521-t003:** Atomization energies per bond (in kcal/mol) of the six molecules in the AE6 dataset.

	Active Space	CASPT2	VBSCF	λ-DFVB		BLYP	B3LYP	Ref. ^1^
				K	IS			
SiH_4_	(8, 8)	79.3	73.4	79.7	79.1	79.1	80.5	81.1
S_2_	(8, 6)	98.0	76.0	92.8	93.8	104.8	100.7	103.1
SiO	(6, 6)	184.6	190.3	193.3	190.1	192.7	184.8	192.4
C_3_H_4_	(8, 8)	116.4	101.5	115.0	114.5	116.9	117.0	117.5
C_2_H_2_O_2_	(10, 10)	124.7	108.0	125.3	123.9	128.2	126.1	126.7
C_4_H_8_	(8, 8)	94.4	77.7	93.2	92.8	94.3	95.2	95.8
	MUE	3.2	14.9	3.1	3.7	1.2	2.0	

^1^ The reference values are taken from [75].

**Table 4 molecules-26-00521-t004:** The excitation energies (in eV) of several atoms in the second row.

	Excitation	Active Space	CASPT2	VBSCF	λ-DFVB		BLYP	B3LYP	Ref. ^1^
					K	IS			
Be	^1^S→^3^P	(2, 4)	2.78	2.81	3.66	3.08	2.48	2.46	2.73
C	^3^P→^1^D	(4, 4)	1.26	1.53	1.14	1.19	0.33	0.38	1.26
N^+^	^3^P→^1^D	(4, 4)	1.87	2.12	1.72	1.79	0.56	0.62	1.89
N	^4^S→^2^D	(5, 4)	2.47	2.79	2.13	2.14	0.94	1.04	2.38
O^+^	^4^S→^2^D	(5, 4)	3.40	3.70	3.03	3.03	1.4	1.53	3.32
O	^3^P→^1^D	(6, 4)	1.91	2.18	1.78	1.81	0.65	0.70	1.96
	MUE		0.05	0.27	0.32	0.20	1.20	1.14	

^1^ The reference values of C, N^+^, N, and O are taken from [76], and those of Be and O^+^ are from [77,78], respectively.

**Table 5 molecules-26-00521-t005:** Forward and reverse barrier heights (in kcal/mol) of the twelve reactions in the DBH24 dataset.

	Active Space	CASPT2	VBSCF	λ-DFVB		BLYP	B3LYP	Ref. ^1^
				K	IS			
OH + CH_4_ → CH_3_ + H_2_O	(3, 3)	5.9	23.9	3.8	2.2	−2.3	2.3	6.3
reverse		19.6	29.6	23.1	25.8	10.4	13.8	19.5
H + OH → O + H_2_	(4, 4)	11.7	17.9	11.0	11.4	1.3	3.9	10.9
reverse		14.2	29.4	14.8	15.4	1.5	6.3	13.2
H + H_2_S → HS + H_2_	(3, 3)	5.1	11.8	4.2	3.1	−2.3	−0.7	3.9
reverse		18.3	26.1	21.3	24.2	14.7	16.2	17.2
H + N_2_O → N_2_ + OH	(11, 9)	19.7	30.8	18.7	18.8	8.7	11.5	17.7
reverse		81.2	108.7	78.7	75.9	62.4	73.5	82.6
H + ClH → HCl + H	(3, 3)	19.7	29.6	18.2	16.8	10.5	13.1	17.8
reverse		19.7	29.6	18.2	16.8	10.5	13.1	17.8
CH_3_ + FCl → CH_3_F + Cl	(3, 3)	6.5	15.0	8.5	14.5	−7.1	−1.6	7.1
reverse		63.7	71.4	76.3	73.9	42.4	51.6	59.8
Cl^−^ ··· CH_3_Cl → ClCH_3_ ··· Cl^−^	(4, 3)	10.0	19.3	10.3	12.7	5.2	8.7	13.5
reverse		10.0	19.3	10.3	12.7	5.2	8.7	13.5
F^−^ ··· CH_3_Cl → FCH3 ··· Cl^−^	(4, 3)	4.2	5.6	1.5	2.4	−1.8	0.2	3.5
reverse		28.2	47.9	31.2	34.6	20.6	26.3	29.6
OH^−^ + CH_3_F → HOCH_3_ + F^−^	(4, 3)	−0.6	8.9	−3.5	0.9	−7.9	−4.5	−2.7
reverse		19.9	28.6	18.5	20.3	11.5	15.9	17.6
H + N_2_ → HN_2_	(7, 7)	17.4	29.0	16.6	16.5	5.4	7.7	14.6
reverse		11.2	−2.0	7.3	6.9	8.5	10.9	10.9
H + C_2_H_4_ → CH_3_CH_2_	(3, 3)	2.7	7.6	3.3	1.7	−0.7	−0.2	2.0
reverse		41.4	37.7	43.6	44.3	38.2	41.8	42.0
HCN → HNC	(10, 9)	48.2	56.3	46.8	44.4	46.8	47.4	48.1
reverse		32.6	36.9	37.0	38.3	31.9	33.5	33.0
MUE		1.4	10.6	2.6	3.5	7.7	4.2	

^1^ The reference values are taken from [75].

## Data Availability

The original data presented in this study are available by request to the authors.

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
