# Peer review of "A Valence-Bond-Based Multiconfigurational Density Functional Theory: The λ-DFVB Method Revisited"

_molecules, 2021, doi:10.3390/molecules26030521_

Round 1
Reviewer 1 Report
The paper presents a modification of the combined VB and DFT model in ref. 37, where the coupling parameter is chosen based on natural orbitals. The ref 37 model in turn is a VB analogue of the MC-DFT approach originally proposed by Savin. The new procedure amounts to a posterori correction to the VBSCF energy. The new version is shown to perform at par or better than the previous method for a selection of properties for small systems.
The authors state that KS-DFT has problems with systems having strong multi-reference character due to the use of a single determinant. This is not true. It has been shown that any density derived from a multi-determinantal wave function, including full-CI, can be exactly reproduced by a set of orbitals in a single determinant (e.g. https://doi.org/10.1021/acs.jctc.0c01029 and references therein). The KS-DFT problem is thus that none of the current XC functional are well suited for describing strong correlation, but that is not a failure of KS-DFT itself.
The authors should also mention the on-top pair density approach for describing strong correlation.
It could have been of interest to compare the VB-DFT with the corresponding MC-DFT approach, rather than CASPT2, as they should show very similar performance. I suggest the authors comment on where they see the VB-DFT approach possibly could be preferred over MC-DFT, as the latter presumably is computationally more efficient.
Author Response
The paper presents a modification of the combined VB and DFT model in ref. 37, where the coupling parameter is chosen based on natural orbitals. The ref 37 model in turn is a VB analogue of the MC-DFT approach originally proposed by Savin. The new procedure amounts to a posterori correction to the VBSCF energy. The new version is shown to perform at par or better than the previous method for a selection of properties for small systems.
(1) The authors state that KS-DFT has problems with systems having strong multi-reference character due to the use of a single determinant. This is not true. It has been shown that any density derived from a multi-determinantal wave function, including full-CI, can be exactly reproduced by a set of orbitals in a single determinant (e.g. https://doi.org/10.1021/acs.jctc.0c01029 and references therein). The KS-DFT problem is thus that none of the current XC functional are well suited for describing strong correlation, but that is not a failure of KS-DFT itself.
(1) Reply: Thanks for the comments, we agree with the reviewer’s comment. We revised our statements in the third paragraph of introduction, which is shown as “…Due to the fact that the exact formula of exchange-correlation functional is still unknown, KS-DFT currently suffers from a difficulty in tackling molecular systems with strong multireference character, which can not be described well by using single Slater determinant.”.
(2) The authors should also mention the on-top pair density approach for describing strong correlation.
(2) Reply: Thanks for the reviewer’s suggestion, we have added the comments for the on-top pair density approach in the third paragraph of introduction, which is “…By using on-top pair density, various DCE-free MRDFT methods have been proposed for the last two decades. Among them, one of the representative MRDFT methods is the multiconfiguration pair-density functional theory (MC-PDFT), which avoids the DCE by calculating the kinetic and Coulomb contributions from the wave function part and the rest part is obtained from a pair-density functional [25].”
(3) It could have been of interest to compare the VB-DFT with the corresponding MC-DFT approach, rather than CASPT2, as they should show very similar performance. I suggest the authors comment on where they see the VB-DFT approach possibly could be preferred over MC-DFT, as the latter presumably is computationally more efficient.
(3) Reply: Thanks for the advice. We have added the comparisons of the computational results between our method and MC-PDFT in the discussions for the atomization energies and chemical reaction barriers. Their performances are similar.
Reviewer 2 Report
The work is very interesting, providing a new relatively precise, efficient, and robust multireference λ-DFVB(IS) approach for the calculation of a broad amount of properties of energetically quasi-degenerate molecular structures.
The major methodological novelty explored by the authors in the MS is an appropriate balanced and optimized combination of the multireference valence bond method and density functional theory via the parameterization based on natural orbital occupation numbers.
The good performance of λ-DFVB(IS) was demonstrated on a broad range of
molecular properties, including equilibrium bond lengths and dissociation energies, atomization energies, atomic excitation energies, and chemical reaction barriers. Computational results show that λ-DFVB(IS) is more robust and close in accuracy to the high-level multireference wave function method, such as CASPT2.
I think that the MS is of high interest and importance to many researchers in molecular physics. It is a well-focused and well-written paper and could be published in Molecules in the present form.
Author Response
The work is very interesting, providing a new relatively precise, efficient, and robust multireference λ-DFVB(IS) approach for the calculation of a broad amount of properties of energetically quasi-degenerate molecular structures. The major methodological novelty explored by the authors in the MS is an appropriate balanced and optimized combination of the multireference valence bond method and density functional theory via the parameterization based on natural orbital occupation numbers. The good performance of λ-DFVB(IS) was demonstrated on a broad range of molecular properties, including equilibrium bond lengths and dissociation energies, atomization energies, atomic excitation energies, and chemical reaction barriers. Computational results show that λ-DFVB(IS) is more robust and close in accuracy to the high-level multireference wave function method, such as CASPT2. I think that the MS is of high interest and importance to many researchers in molecular physics. It is a well-focused and well-written paper and could be published in Molecules in the present form. Reply: We sincerely thank the reviewer for the positive comments.